# Gram Stain and Culture of Sputum Samples Detect Only Few Pathogens in Community-Acquired Lower Respiratory Tract Infections: Secondary Analysis of a Randomized Controlled Trial

**DOI:** 10.3390/diagnostics13040628

**Published:** 2023-02-08

**Authors:** Mariana B. Cartuliares, Helene Skjøt-Arkil, Christian B. Mogensen, Thor A. Skovsted, Steen L. Andersen, Andreas K. Pedersen, Flemming S. Rosenvinge

**Affiliations:** 1Emergency Department, University Hospital of Southern Denmark, 6200 Aabenraa, Denmark; 2Department of Regional Health Research, University of Southern Denmark, 6200 Aabenraa, Denmark; 3Department of Biochemistry and Immunology, University Hospital of Southern Denmark, 6200 Aabenraa, Denmark; 4Department of Clinical Microbiology, University Hospital of Southern Denmark, 6200 Aabenraa, Denmark; 5Department of Clinical Research, University Hospital of Southern Denmark, 6200 Aabenraa, Denmark; 6Department of Clinical Microbiology, Odense University Hospital, 5000 Odense, Denmark; 7Research Unit of Clinical Microbiology, University of Southern Denmark, 5000 Odense, Denmark

**Keywords:** lower respiratory tract infection, emergency department, sputum, Gram stain, culture, microbiology, tracheal suction, expiratory technique, antibiotic treatment

## Abstract

Identification of the bacterial etiology of lower respiratory tract infections (LRTI) is crucial to ensure a narrow-spectrum, targeted antibiotic treatment. However, Gram stain and culture results are often difficult to interpret as they depend strongly on sputum sample quality. We aimed to investigate the diagnostic yield of Gram stain and culture from respiratory samples collected by tracheal suction and expiratory technique from adults admitted with suspected community-acquired LRTI (CA-LRTI). In this secondary analysis of a randomized controlled trial, 177 (62%) samples were collected by tracheal suction, and 108 (38%) by expiratory technique. We detected few pathogenic microorganisms, and regardless of sputum quality, there were no significant differences between the sample types. Common pathogens of CA-LRTI were identified by culture in 19 (7%) samples, with a significant difference between patients with or without prior antibiotic treatment (*p* = 0.007). The clinical value of sputum Gram stain and culture in CA-LRTI is therefore questionable, especially in patients treated with antibiotics.

## 1. Introduction

The diagnostic value of sputum samples in lower respiratory tract infection (LRTI) has been questioned for many years. In clinical practice, microbiological analysis of sputum by Gram stain and culture can determine the etiological agent of LRTI and enable targeted antibiotic treatment [1,2,3]. 

Sputum samples can be collected by tracheal suctioning, induction with saline inhalation, self-expectoration using expiratory techniques, or other methods [4,5,6,7]. A randomized controlled trial (RCT) identified tracheal suction (TS) as the best method to obtain sputum samples of good quality compared to forced expiratory technique combined with induced sputum (FETIS) [8]. This leaves a question of whether good quality sputum samples obtained by TS are better than FETIS for detecting pathogenic microorganisms by Gram stain and culture.

American clinical guidelines recommend that laboratories only culture samples of acceptable quality [9]. Gram stain is used to assess sputum quality, but there is no gold standard and different criteria and thresholds have been suggested. Commonly, quality is defined by the number of squamous epithelial cells (SEC) indicating contamination by oropharyngeal microbiota and polymorphonuclear leukocytes (PMNL) indicating inflammation. In addition, some studies assess quality by calculating the ratio of PMNL/SEC [10,11,12,13,14,15,16].

Community-acquired LRTI (CA-LRTI) is usually caused by common pathogens such as *Streptococcus pneumoniae* and *Haemophilus influenzae*. Enteric and non-fermenting Gram-negative bacilli are notable pathogens in selected patients but are infrequent causes of CA-LRTI, and detection often reflects upper airway colonization [17]. Gram stain is reported to be highly specific for diagnosing *S.pneumoniae*, *H. influenzae*, and other Gram-negative bacilli and can therefore contribute to clinical decisions before pathogens are verified by culture [2,18]. However, the reliability of sputum analysis decreases if the patient has been treated with antibiotics before admission, reducing the clinical usefulness of the results [13,19,20]. 

It is unknown and therefore important to investigate how sample type (TS and FETIS), different sputum quality criteria, and recent antibiotic treatment affect the detection of pathogenic bacteria in CA-LRTI.

We hypothesized that good quality sputum samples collected by TS would detect more potential pathogens of CA-LRTI than samples collected by FETIS. The objectives were (i) to investigate if there was a difference in microorganisms detected by Gram stain and culture from good quality sputum samples obtained by TS and FETIS and (ii) to investigate the impact of prior antibiotic treatment on sputum culture results. 

## 2. Materials and Methods

This study is a secondary analysis of a RCT, and for detailed information about the primary trial, we refer to the statistical analysis plan [21] and the primary study [8]. The trial was conducted from 9 November 2020 to 5 July 2021 at 2 Danish emergency departments (EDs) at Hospital Sønderjylland, with a catchment area of approximately 225,000 inhabitants. Microbiological analyses were performed at the hospital’s Department of Clinical Microbiology. 

Processing of personal data was approved by the Region of Southern Denmark (20/41767) in accordance with the EU General Data Protection Regulation. Furthermore, the study was registered by clinicaltrials.org (NCT04595526 20 October 2020 and completed 5 July 2021) and was approved by the Regional Committee for Health Research Ethics in Southern Denmark (S-20200133).

### 2.1. Participants

Adults (>18 years of age) admitted to the ED with suspected CA-LRTI were invited by ED project assistants and enrolled in the study, if they gave verbal and written consent, and if the attending physician identified at least one of the following pulmonary symptoms: dyspnea, cough, expectoration, chest pain, or fever. Patients were excluded if participation delayed urgent treatment, transfer to an intensive care unit, or if the patient had severe immunodeficiency [21].

### 2.2. Randomization

Patients were randomly assigned (1:1) to either tracheal suction (TS) (usual care) or forced expiratory technique (FET) combined with sputum induction (FETIS) (intervention). Randomization was computer generated [22] and performed by the project assistants before collecting the sputum sample [21].

### 2.3. Sampling Methods

Respiratory samples (tracheal secretions and expectorated sputa) were collected after the initial clinical assessment or within 24 h of admission. TS was performed according to local guidelines. The patient was placed in Fowler’s position and encouraged to clear the airways with a deep cough. The suction catheter (EXTRUDAN Surgery Aps, Denmark, CH12, 530 mm) tip was lubricated with Xylocaine (lidocaine HCl) 2% jelly, inserted into the nares during inhalation, and gently advanced about 40 cm without applying suction. Suction was performed at 200–400 mmHg negative pressure before withdrawing the catheter. FETIS was performed according to a standardized protocol [21] and was based on the patients’ attempts to deliver a sputum sample. It included ng FET alone and induced sputum (IS) combined with FET [5,23]. The patient was placed in a 90° sitting position, the mouth was cleared with water to minimize oropharyngeal contamination, and the sample was obtained by forced exhalation and coughing [5]. Using the same procedure, a second sputum sample was obtained after inhalation of nebulized isotonic saline (Unomedical Opti-Mist TM, 2.1 m, ref. 93–772 mm) [23]. Hence, each patient in the intervention group (FETIS) could deliver two samples. Participants in the intervention group who could not deliver a sputum sample by FETIS underwent tracheal suction (TS-IG); these samples were also included in the secondary analysis [21].

### 2.4. Gram Stain and Culture

A part of the sputum specimen was placed on a microscope slide with a cotton swab, and a second slide was used to distribute the material on the surface. The smear was then heat fixed and Gram stained. For each sputum sample, the number of SEC and PMNL per field of view (10× objective) were recorded. Sputum samples were classified as good quality by three different criteria: (i) <10 SEC, (ii) <10 SEC and >25 PMNL, or (iii) PMNL/SEC ratio > 5. Samples with SEC < 10 and PMNL > 25 were defined as purulent.

Microorganisms were classified as Gram-positive or Gram-negative and by morphology (rods, cocci (pairs, chains, clusters), yeast) (×100 objective).

The remaining part of the sputum sample was transferred to a 5% sheep blood agar plate (Beckton Dickinson, BD, Sparks, MD, USA) and a Chrom Orientation agar plate (BD) and streaked over the agar surface with a sterile inoculation loop. Blood agar plates were inoculated with a *Staphylococcus aureus* streak to allow growth of *H. influenzae*. Agar plates were incubated at 35 °C in normal atmospheric conditions (Chrom-agar Orientation) and in a 5% CO_2_ atmosphere (5% blood agar).

After 1–2 days of incubation, pathogens were identified by matrix-assisted laser desorption/ionization–time of flight. In addition, “no growth of pathogens” and “upper airway microbiota” were reported. If culture and microscopy were incongruous, microscopy was re-evaluated, and the agar plates were incubated for two more days.

Microscopy and culture results were registered in the microbiological laboratory information system (MADS, Aarhus University Hospital, Aarhus, Denmark) and were accessible from the patient’s medical chart. 

### 2.5. Group of Pathogens

Detected microorganisms were classified into four groups:(1).Common pathogens of CA-LRTI (*S. pneumoniae*, *H. influenzae*, and *Moraxella catarrhalis*);(2).Possible pathogens of CA-LRTI (*Pseudomonas aeruginosa* and *S. aureus*);(3).Unlikely pathogens of CA-LRTI (*Enterobacterales*, *Enterococcus* sp., *Neisseria meningitidis*, *S. maltophila*, *Streptococcus agalactiae*, and yeast);(4).Upper airway microbiota.

Respiratory pathogens classified as ‘common pathogens’ represent the most predominant etiologies of CA-LRTI [24,25,26,27,28]. ‘Possible pathogens’ may cause CA-LRTI, especially in patients with underlying respiratory diseases, but more often represent colonization [29]. ‘Unlikely pathogens’ represents pathogens that rarely cause CA-LRTI and usually originate from upper airway colonization [17,28].

### 2.6. Statistical Analysis

The secondary analyses included specimens collected from patients randomized to TS and FETIS. Tracheal secretions from patients randomized to the FETIS group who could not deliver a sample by self-expectoration and therefore underwent tracheal suction were also included. Hence, two groups were analyzed: TS (included TS-SG from standard care group and TS-IG from intervention group) and FETIS (FET and IS). Sensitivity analyses were performed for the four sampling methods (TS-SG, TS-IG, FET, and IS). 

Fisher’s exact test was performed to compare the yield of microorganisms identified by Gram stain and culture from (1) TS and FETIS and (2) TS and FETIS stratified on the different quality criteria (<10 SEC, <10 SEC and ≥25 PML, >5 PMNL/SEC).

The impact of antibiotic treatment on the yield of pathogens from culture was presented descriptively and using the Chi-Square test based on the defined pathogen groups.

The analysis was based on complete cases and no multiple imputation was performed. A *p*-value less than 0.05 was considered statistically significant, and no adjustments for multiple testing were utilized. Statistical analyses were performed using STATA 17.0 (TX, USA).

## 3. Results

In total, 285 specimens were collected from 280 patients between 10 November 2020 and 5 July 2021. We included 177 (62%) tracheal secretions (120 TS-SG and 57 TS-IG) and 108 (38%) sputum samples (50 FET and 58 IS) (Figure 1).

Gram stains and culture results from samples collected by TS and FETIS are presented in Table 1. Sensitivity analysis from Gram stains and culture results from samples collected by TS and FET and TS and IS are presented in the Appendix A (Appendix A).

Gram stain detected possible pathogens in 59 samples (21%). A total of 2 or more different possible pathogens were detected by Gram stain in 16 (6%) samples: 8 (6%) from TS and 8 (7%) from FETIS. There was no significant difference between detected pathogens from TS and FETIS (*p* = 0.384). TS samples were less contaminated with upper airway microbiota 57 (32%) compared to FETIS 60 (55%) (*p* = 0.001).

We identified microorganisms by culture in 105 samples (37%). In 15 (6%) samples, more than 1 microorganism was identified. Overall, there was no significant difference between culture results when comparing TS and FETIS in relation to identified microorganisms (*p* = 0.325), “upper airway microbiota” (*p* = 0.765), and “no growth of pathogens” (*p* = 0.174).

Culture results for TS and FETIS stratified on the three different quality criteria are presented in Table 2. 

In good quality sputa (<10 SEC), we identified more microorganisms in TS samples (49 (28%)) than in FETIS samples (23 (21%)); however, the difference between the groups was not statistically significant (*p* = 0.096). In purulent samples and in samples with a PMNL/SEC ratio ≥ 5, culture results from TS and FETIS were similar (*p* = 0.955 and *p* = 0.457, respectively). Results stratified in the three quality criteria showed no statistical difference between TS and FETIS in relation to “upper airway microbiota” (*p* = 0.561, *p* = 0.500, *p* = 0.195) and “no growth of pathogens” (*p* = 0.053, *p* = 0.306, *p* = 0.124). 

The 260 culture results were categorized as ‘Common pathogens of CA-LRTI’ [19 (7.3%): *H. influenzae* (7 (2.7%)), *S. pneumoniae* (6 (2.3%)), *M. catarrhalis* (6 (2.3%))], ‘Possible pathogens of CA-LRTI’ [25 (9.6%): *S. aureus* (23 (8.8%)), *P. aeruginosa* (2 (0.8%))], ‘Unlikely pathogens of CA-LRTI’ [76 (29.2%): *Enterobacterales* (45 (17.3%)), yeast (26 (10%)), other (5 (2%))], ‘Upper airway microbiota’ (42 (16%)), and ‘No growth of pathogens’ (98 (37.7%)). The associations between the pathogen group, sampling method, Gram stain quality criteria, and prior antibiotic treatment are shown in Figure 2. Most samples of good quality (<10 SEC) were obtained by TS and ‘Common pathogens of CA-LRT’ were most often identified in good quality samples (14 (5.4%)), where 8 (3%) were from purulent samples (<10 SEC and >25 PMNL). Most purulent samples (2/3) were obtained by FETIS.

Compared to patients without prior antibiotic treatment, samples from patients treated with antibiotics within one month before admission yielded significantly fewer common pathogens (*p* = 0.007), possible pathogens (*p* = 0.018), and significantly more unlikely pathogens (*p* < 0.001) of CA-LRTI (Table 3). Sensitivity analysis of detected pathogens (common, possible, and unlikely) stratified by sampling methods (TS and FET, TS and IS) did not change the overall results and are presented in the Appendix A (Appendix A). Identified microorganisms from patients untreated and treated with antibiotics are presented in the Appendix A (Appendix A).

## 4. Discussion

In this study, we described the Gram stain and culture findings in 285 sputum samples collected by TS and FETIS from CA-LRTI patients. Regardless of different quality criteria, there was no statistically significant difference in culture results between TS and FETIS. Samples obtained by TS were assessed to be less contaminated with upper airway microbiota by Gram stain. This result indicates that TS is better than FETIS for obtaining samples from the lower airways. Few common pathogens of CA-LRTI (*H. influenzae, S. pneumoniae,* and *M. catarrhalis*) were identified by culture. Not surprisingly, samples from patients not treated with antibiotics before admission yielded almost twice as many common pathogens of CA-LRTI compared to samples from patients treated with antibiotics. 

A retrospective multicenter study from 2022 reported a different result, concluding that the diagnostic yield was higher for expectorated and induced sputum compared to tracheal secretions [24]. However, the study included very few patients in the TS group (21 (1.6%)), indicating sample bias. 

We identified common pathogens of CA-LRTI in 19 (7.3%) of the 260 culture results—almost all 14 (5.4%) from good quality samples with <10 SEC and over half were from purulent samples (<10 SEC and >25 PMNL). These findings, in accordance with earlier observations, showed that good quality sputum with <10 SEC and >25 PMNL was 3.8 times more likely to grow pathogenic bacteria compared to poor quality sputum [30]. A recent systematic review found an increased diagnostic yield of the Gram stain in identifying bacterial etiologies of CAP when samples of good quality were obtained [2]. It indicates that quality classification by Gram stain is important and contributes to accurate diagnostics of CAP pathogens.

No pathogen was detected by culture in 140 (49%) samples. There are several possible explanations for the low yield of the Gram stain and culture. An explanation could be that the pathogens of LRTI generally are difficult to detect. A study with a high level of patient participation (95%) failed to determine the etiology for 47% of the patients [26]. Meanwhile, another study comparing paired sputa and transtracheal aspirated samples revealed that if a specimen of good quality (<10 SEC and >25 PMNL) did not identify a pathogen, there was still a 45% chance that a pathogen was detected in the paired transtracheal aspirate [10]. These findings suggest that bacterial culture has a low sensitivity in detecting causative pathogens of CA-LRTI.

Patients were enrolled in this study based on clinical symptoms (dyspnea, cough, expectoration, chest pain, or fever), and before results from chest X-rays, blood tests, and urine tests were available. These symptoms are common in patients with bacterial CA-LRTI but also in patients with viral infections [31]. Therefore, another explanation for the low number of culture-positive sputa could be that a high number of patients in our study was admitted with viral infections. Unfortunately, the only registered viral agent was SARS-CoV-2 as data were prospectively collected with pre-specified variables. Forty (14%) patients were infected with SARS-CoV-2 [8]. On the other hand, surveillance data show that there was a very low transmission in Denmark of other common respiratory viruses during the SARS-CoV-2 pandemic [29]. This might have reduced the number of admissions caused by viral infections and may also have reduced the risk of secondary bacterial pneumonia—possibly in part explaining the low detection of bacterial pathogens in our study.

Finally, another probable explanation is that patients with chronic obstructive pulmonary disease (COPD) might have been included in the study with acute exacerbation rather than CA-LRTI as they are often admitted with similar clinical symptoms, e.g., increased sputum production, sputum purulence, and dyspnea. There is an etiological overlap, but acute exacerbations in COPD are often caused by non-bacterial etiologies (e.g., viral infections and environmental factors) [27,28,32].

In line with the literature, many samples in our study, 157 (55%), were analyzed from patients receiving antibiotics within one month before admission [26]. It is well recognized that consumption of antibiotics decreases the diagnostic yield of Gram stain and culture [13,19,33]. A previous study reported an association between prior antibiotic treatment and a four-fold reduction in diagnostic yield [33]. Furthermore, the detection of causative pathogens of LRTI is significantly reduced if antibiotics are consumed within 24 h before collecting a sputum sample [13,19]. Our study correlates well with these studies, with a statistically significant difference in culture results from patients with or without previous antibiotic treatment. Our study also supports the observation from other studies that more unlikely pathogens of CA-LRTI, such as *Enterobacterales*, are detected in patients previously treated with antibiotics [13,19]. *Enterobacterales* rarely cause pneumonia in a community setting and the detection is probably a result of antibiotic selective pressure and oropharyngeal overgrowth [25]. Albeit an infrequent cause, the bacteria may cause severe LRTI, highlighting the importance of separating etiology and colonization [17]. Antibiotic therapy may also explain the observation that in this study almost half of TS samples, despite good quality, had no growth of pathogens. The question remaining is: “Do samples from patients previously treated with antibiotics add value to clinical practice?”

The major strengths of this study are the high rate of obtained specimens (88%) from patients with suspected CA-LRTI and the randomized prospective design of the primary study minimizing sampling bias. In addition, samples were collected, Gram stained, and cultured by standardized and closely monitored procedures. A major limitation is, similar to other studies, the low number of microorganisms identified, especially common pathogens of CA-LRTI. If significantly more samples had been included, we may have detected a difference in diagnostic yield, but the study size was fixed by the primary study [8]. Another limitation was not including patient discharge diagnosis, viral test results, and blood culture results, which may have confirmed or supplemented our results. Many samples were from patients who received antibiotics before admission (55%). However, this is not regarded as a limitation as this prospective study gives insights into real-world practice and challenges in managing acutely admitted patients with suspected CA-LRTI.

## 5. Conclusions

We detected very few relevant pathogens of CA-LRTI regardless of sample type (TS/FETIS), sample quality, and microbiological test (Gram stain/culture), especially in patients treated with antibiotics. Future research should focus on methods to mitigate this problem. It is possible that molecular methods, e.g., syndromic test panels, will have a higher diagnostic sensitivity, will be less sensitive to prior antibiotic treatment, and in addition, allow the detection of both viral and bacterial pathogens. Regardless of the method, it will still be essential to ensure specimens of optimal quality as most bacterial pathogens also are commensals of the upper airways.

## Figures and Tables

**Figure 1 diagnostics-13-00628-f001:**
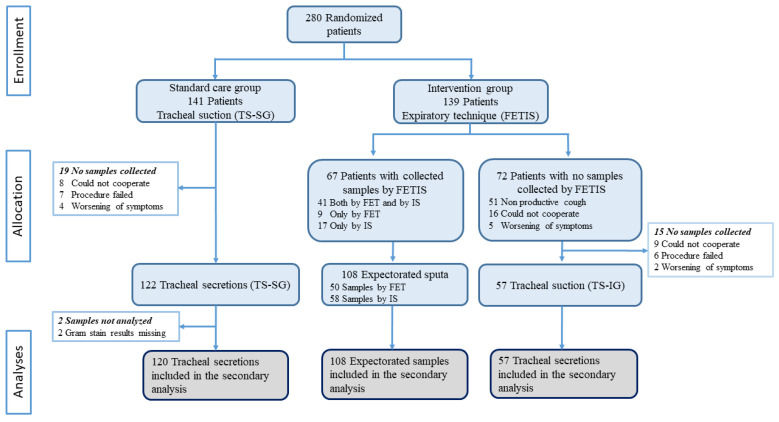
Trial profile and description of the collection of samples. FET: forced expiratory technique; IS: induced sputum; FETIS: forced expiratory technique and induced sputum; TS-SG: tracheal suction standard care group; TS-IG: tracheal suction intervention group.

**Figure 2 diagnostics-13-00628-f002:**
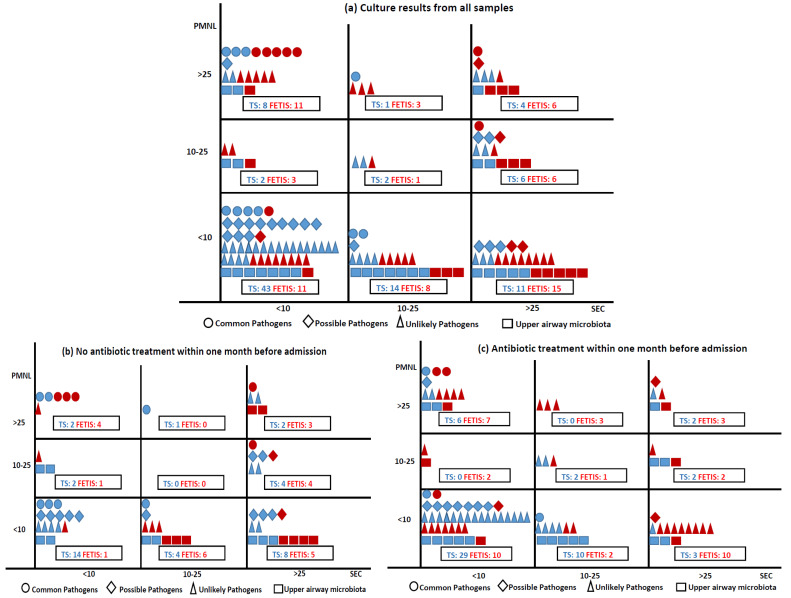
Samples collected by TS and FETIS distributed according to Gram stain quality criteria, culture results (**a**), and samples without and with prior antibiotic treatment (**b**,**c**). Sampling methods: TS blue, FETIS red. Culture results are presented in four groups: ○: common pathogens of CA-LRTI (*S. pneumoniae*, *H. influenzae*, and *M. catarrhalis*), ◊: possible pathogens of CA-LRTI (*P. aeruginosa* and *S. aureus*), ∆: unlikely pathogens of CA-LRTI (*Enterobacterales*, *Enterococcus* sp., *N. meningitidis*, *S. maltophila*, *S. agalactiae*, and yeast), and □: upper airway microbiota. A total of 9 culture results were missing: ○ 1 FETIS (<10 SEC), ◊ 1 TS (<10 SEC), ∆ 6 TS (<10 SEC), □ 1 TS (<10 SEC).

**Table 1 diagnostics-13-00628-t001:** Findings from Gram stain and culture from tracheal secretions (TS) and sputum samples collected by FET and IS (FETIS).

		Sampling Method		Total	*p*-Value
Total *n* (%)		TS 177 (62%)	FETIS 108 (38%)	*n* = 285	
**Gram stain**					
	Number of positive samples	31 (18%)	28 (26%)	59 (21%) *	
	All potential pathogens	39 (22%)	36 (33%)	75 (26%) *	0.384
	Gram-positive cocci chains/pairs	15 (38%)	10 (28%)	25 (33%) ^#^	
	Gram-positive cocci clusters	3 (8%)	3 (8%)	6 (8%) ^#^	
	Gram-negative rods	5 (13%)	12 (33%)	17 (23%) ^#^	
	Gram-positive rods	2 (5%)	1 (3%)	3 (4%) ^#^	
	Gram-positive single	5 (13%)	4 (11%)	9 (12%) ^#^	
	Gram-negative diplococci	3 (8%)	4 (11%)	7 (9%) ^#^	
	Yeast	6 (15%)	2 (6%)	8 (11%) ^#^	
	Upper airway microbiota	57 (32%)	60 (56%)	117 (41%) *	0.001
**Culture**					
	Number of positive samples *	63 (36%)	42 (39%)	105 (37%) *	
	All potential pathogens *	72 (41%)	48 (44%)	120 (42%) *	0.325
	*Streptococcus pneumoniae*	4 (6%)	2 (4%)	6 (5%) ^#^	
	*Enterococcus* sp.	2 (3%)	0 (0%)	2 (1%) ^#^	
	*Staphylococcus aureus*	19 (26%)	4 (8%)	23 (19%) ^#^	
	*Haemophilus influenzae*	3 (4%)	4 (8%)	7 (6%) ^#^	
	*Enterobacterales*	25 (35%)	20 (42%)	45 (38%) ^#^	
	*Moraxella catarrhalis*	3 (4%)	3 (6%)	6 (5%) ^#^	
	*Pseudomonas aeruginosa*	1 (1%)	1 (2%)	2 (1%) ^#^	
	Other	1 (1%)	2 (4%)	3 (3%) ^#^	
	Yeast	14 (19%)	12 (25%)	26 (22%) ^#^	
	Upper airway microbiota *	27 (15%)	15 (14%)	42 (15%) *	0.765
	No growth of pathogens	67 (38%)	31 (29%)	98 (34%) *	0.174

* percentage of total; ^#^ percentage of all potential pathogens.

**Table 2 diagnostics-13-00628-t002:** Culture results for TS and FETIS stratified on different quality criteria.

Quality Criteria		Sampling Method		Total	*p*-Value
Total *n* (%)		TS 177 (62%)	FETIS 108 (38%)	*n* = 285	
**<10 SEC ^†^**					
	Number of positive samples	45 (25%)	21 (19%)	66 (23%) *	
	All potential pathogens	49 (28%)	23 (21%)	72 (25%) *	0.096
	*Streptococcus pneumoniae*	3 (6%)	2 (9%)	5 (7%) ^#^	
	*Haemophilus influenzae*	1 (2%)	2 (9%)	3 (4%) ^#^	
	*Pseudomonas aeruginosa*	1 (2%)	0 (0%)	1 (1%) ^#^	
	*Moraxella catarrhalis*	3 (6%)	3 (13%)	6 (8%) ^#^	
	*Staphylococcus aureus*	13 (27%)	1 (4%)	14 (19%) ^#^	
	*Enterobacterales*	17 (35%)	8 (34%)	25 (35%) ^#^	
	*Enterococcus* sp.	1 (2%)	0 (0%)	1 (1%) ^#^	
	Yeast	10 (20%)	5 (22%)	15 (21%) ^#^	
	Other *	0 (0%)	2 (9%)	2 (3%) ^#^	
	Upper airway microbiota	11 (6%)	3 (3%)	14 (5%) *	0.561
	No growth of pathogens	43 (24%)	7 (6%)	50 (17%) *	0.053
**<10 SEC and >25 PMNL ^††^**					
	Number of positive samples	5 (3%)	8 (7%)	13 (5%) *	
	All potential pathogens	6 (3%)	10 (9%)	16 (6%) *	0.955
	*Streptococcus pneumoniae*	1 (17%)	1 (10%)	2 (12%) ^#^	
	*Haemophilus influenzae*	1 (17%)	2 (20%)	3 (19%) ^#^	
	*Moraxella catarrhalis*	1 (17%)	2 (20%)	3 (19%) ^#^	
	*Staphylococcus aureus*	1 (17%)	0 (0%)	1 (6%) ^#^	
	*Enterobacteriaceae*	2 (33%)	4 (40%)	6 (38%) ^#^	
	Other **	0 (0%)	1 (10%)	1 (6%) ^#^	
	Upper airway microbiota	2 (1%)	1 (1%)	3 (1%) *	0.500
	No growth of pathogens	3 (2%)	1 (1%)	4 (1%) *	0.306
**≥5 PMNL/SEC**					
	Number ofpositive samples	9 (5%)	9 (8%)	18 (6%) *	
	All potential pathogens	11 (6%)	11 (10%)	22 (8%) *	0.457
	*Streptococcus pneumoniae*	1 (9%)	1 (9%)	2 (9%) ^#^	
	*Haemophilus influenzae*	1 (9%)	2 (18%)	3 (14%) ^#^	
	*Moraxella catarrhalis*	1 (9%)	2 (18%)	3 (14%) ^#^	
	*Staphylococcus aureus*	3 (27%)	0 (0%)	3 (14%) ^#^	
	*Enterobacterales*	4 (36%)	4 (36%)	8 (36%) ^#^	
	Yeast	1 (9%)	0 (0%)	1 (4%) ^#^	
	Other ***	0 (0%)	2 (18%)	2 (9%) ^#^	
	Upper airway microbiota	1 (<1%)	3 (3%)	4 (1%) *	0.195
	No growth of pathogens	10 (6%)	3 (3%)	13 (5%) *	0.124

* percentage of total; ^#^ percentage of all potential pathogens; ^†^ SEC: squamous epithelial cells; ^††^ PMNL: polymorph nuclear leucocytes; * Other: *N. meningitidis* and *S. maltophila*; ** Other: *S. maltophila*; *** Other: *N. meningitidis* and *S. maltophila*. A total of 43 and 44 samples were missing from the <10 SEC and >25 PMNL and ≥PMNL/SEC group, respectively.

**Table 3 diagnostics-13-00628-t003:** Detected pathogens: relation to antibiotic treatment within one month before admission.

	Antibiotics (NO)	Antibiotic (YES)	Total	*p*-Value
	*n* = 128 (45%)	*n* = 157 (55%)	*n* = 285	
Number of positive samples	40 (31%)	65 (41%)	105 (37%) *	
Common pathogens of CA-LRTI	12 (9%)	7 (4%)	19 (7%) *	0.007
Possible pathogens of CA-LRTI	14 (11%)	11 (7%)	25 (9%) *	0.018
Unlikely pathogens of CA-LRTI	17 (13%)	59 (38%)	76 (27%) *	<0.001
Upper airway microbiota	18 (14%)	24 (15%)	42 (15%) *	0.784

* percentage of total. Culture results of four groups of pathogens: (1) common pathogens of CA-LRT (*S. pneumoniae, H. influenzae,* and *M. catarrhalis*), (2) possible pathogens of CA-LRTI (*P. aeruginosa* and *S. aureus*), (3) unlikely pathogens of CA-LRT (*Enterobacterales*, *Enterococcus* sp. *N. meningitidis*, *S. maltophila*, *S. agalactiae*, and yeast).

## Data Availability

Due to Danish laws on personal data, data cannot be shared publicly. To request data, please contact the corresponding author for more information. The person responsible for the research was the principal investigator and corresponding author (MBC) in collaboration with the Department of Health Research and the University Hospital of Southern Denmark. These organizations own the data and can provide access to the final data set.

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
