# Peer review of "Gram Stain and Culture of Sputum Samples Detect Only Few Pathogens in Community-Acquired Lower Respiratory Tract Infections: Secondary Analysis of a Randomized Controlled Trial"

_diagnostics, 2023, doi:10.3390/diagnostics13040628_

Round 1

Reviewer 1 Report

Thank you for the opportunity of reviewing this manuscript. I have several comments/suggestions:

1. In Section 2.3 (Sampling Methods), it may be helpful to describe the sample collection methods for TS and FETIS to aid the reader, space permitting.

2. Throughout the study, it is noted that 41 patients submitted two sputum samples each - one by FET alone and one by IS. I am concerned that this will lead to bias within the FETIS group as these 41 patients are essentially being counted twice. Do the authors have a proposal for overcoming this?

3. Line 177 states 'two or more possible pathogens were detected by Gram stain in 19 samples...' and Line 182 states 'In 15 samples more than one microorganism was identified'. However, I could not reconcile these statements with the data presented in the tables. For example, in Table 1, Gram stain section, 'all potential pathogens' = 75, and the sum of the Gram stain categories (25 + 6 +17 + 3 + 9 + 7 + 8) is also 75. If 19 samples had two more more pathogens on Gram stain, we would expect the total number of 'positive' samples to be less than 75. This may lead to problems with statistical analysis. The same issue is noted in the 'Culture' section of Table 1.

4. Lines 189 and 201 appear to be referring to some missing samples but it is not clear what the numbers mean. For example, line 189 states '112 and 40 samples were missing...' Could the authors clarify what this means?

5. Line 203 states 'in good quality sputa, we identified more microorganisms in TS samples than in FETIS samples', however according to Table 2 the p-value for this comparison is 0.096 which is not statistically significant per the study methods.

6. Line 245 states 'Samples obtained by TS were more likely to be without growth of pathogens...' however according to Table 1 the p-value for this comparison is 0.174 which is not statistically significant.

Reviewer 2 Report

Correct as attached

Author Response

Thank you very much for your time and your comments. We have revised our manuscript accordingly and in addition, the language of the manuscript has been revised by a colleague with a postgraduate degree in Health and English as a first language.

Kind Regards

Mariana Bichuette Cartuliares

Reviewer 3 Report

The manuscript ”Gram stain and culture of sputum samples are of limited diagnostic value: Secondary analysis of a randomized controlled trial” describes an investigation conducted in induced sputum samples compared to samples acquired with tracheal suction to determine if tracheal suction yields a better sample to determine bacterial etiology of lower respiratory tract infections. Techniques to identify lower respiratory tract pathogens were culture, Gram staining and MALDI-TOF. The manuscript describes an interesting problem: What sample should be collected and what analyses should be conducted to acquire the best clinical information? However, the connection between results and conclusions are unclear.

Specifically, the most confusing statements are:

Line 243: Regardless of different quality criteria, there was no statistically significant difference in culture results between tracheal suction (TS) and forced expiratory technique combined with induced sputum (FETIS). Samples obtained by TS were more likely to be without growth of pathogens and were assessed to be less contaminated with upper airway microbiota by Gram stain. Both results indicate that TS is better than FETIS for obtaining lower respiratory samples.

Here, the authors say that there were no differences between the sampling methods, but then they go on to describe differences. The text must be changed to explain how there can be no differences and differences at the same time.

Furthermore, the authors go on to criticize the methods:

The authors conclude that “there were no significant differences in pathogenic microorganisms between the sample types” and then go on to say that “The clinical value of sputum Gram stain and culture is therefore questionable, especially in patients treated with antibiotics”. Do the authors mean that sputum should not be collected as the clinical value is questionable? How should pathogen be determined, if common practice does not yield the results required to determine treatment? If the authors claim this, some other practice should be suggested and motivated.
